# Pathways of emergency care for severely ill children in Nigerian and Ugandan hospitals: A process mapping study

Rami Subhi[1,2*], Abiodun Sogbesan[3,4], Dan Muramuzi[5], Mikael Burhin[1], Ayobami A. Bakare[3,4,6], Adegoke G. Falade[3], Freddy E. Kitutu[5,7,8], Freddie Ssengooba[5], Carina King[4], Sumit Kane[9], Belinda Dawson-McClaren[1‡], Hamish R. Graham[1,3‡], the MOXY-Implementation Research Collaboration[¶]

1 Centre for International Child Health, University of Melbourne, MCRI, Royal Children's Hospital, Melbourne, Victoria, Australia, 2 The Northern Hospital, Melbourne, Victoria, Australia, 3 Oxygen for Life Initiative, Department of Paediatrics, University College Hospital, Ibadan, Nigeria, 4 Department of Global Public Health, Karolinska Institutet, Stockholm, Sweden, 5 Makerere University School of Public Health, Kampala, Uganda, 6 Department of Community Medicine, University College Hospital, Ibadan, Nigeria, 7 Department of Pharmacy, Makerere University School of Health Sciences, Kampala, Uganda, 8 Department of Women's and Children's Health, International Child Health and Migration, Uppsala University, Uppsala, Sweden, 9 Social & Economic Determinants of Health, Melbourne School of Population and Global Health, Melbourne, Victoria, Australia

¶ Membership of the MOXY-Implementation Research Collaboration is provided in the Acknowledgments.
‡ These authors are senior authors on this work.
* rami.subhi@mcri.edu.au

## Abstract

### Background

Child mortality remains high in countries with weak emergency care systems. Facility organisation for paediatric emergency care is heterogeneous and under-described. We examined how hospitals in Uganda and Nigeria are organised to deliver emergency care for neonates and children.

### Methods and findings

We conducted a qualitative, multi-method study in 26 purposively selected secondary and tertiary facilities in Uganda and Nigeria from October 2023 to December 2024. Embedded researchers documented patient pathways, resources for care, and care processes for severely ill children (<15 years). We used inductive content analysis to generate organisational archetypes and describe different facets of the patient journey.

We identified 4 recurring patterns of facility organisation and patient flow ('archetypes'): outpatient department (OPD) 'screen and treat'; OPD 'screen and send'; emergency department (ED) 'receive and treat'; and inpatient department (IPD) 'receive and treat'. Across sites, formal triage systems were generally absent or rarely used. First contact and early sorting of children into these pathways frequently

**Data availability statement:** The full facility process maps cannot be made publicly available because they contain detailed layout schematics that may permit identification of individual facilities, even after removal of explicit identifiers. De-identified facility summaries are provided in the Supporting information files. Requests for access to additional materials may be directed to the relevant country institutional ethics committees (University College Hospital Ibadan/ University College Hospital, Nigeria: uiuchec@ gmail.com; Makerere University School of Health Sciences, Uganda: muwanguzinoreen@ yahoo.com), who can review and, if appropriate, forward requests to facility leadership in accordance with local governance requirements.

**Funding:** This work was funded by a subgrant from the Bill and Melinda Gates Foundation (https://www.gatesfoundation.org/) and ELMA Philanthropies (https://www.elmaphilanthropies. org/) (INV-043011) awarded to HRG, and the Swedish Research Council (https://www.vr.se/ english.html) (reference: 2023-01977), awarded to CK. The funders had no role in study design, data collection and analysis, decision to publish, or preparation of the manuscript.

**Competing interests:** I have read the journal's policy and the authors of this manuscript have the following competing interests: RS and MB receive salary support from the Melbourne Children's Research Institute. HRG was previously Academic Editor for PLOS Global Public Health and was supported with salary support from NHMRC and the RCH Foundation. HRG is a board member of Oxygen for Life Initiative (non-profit) and an advisor for Lifebox and the Global Oxygen Alliance. FEK receives general salary support from Makerere University and holds a sub-grant from the Melbourne Children's Research Institute. AB is a principal officer of Oxygen for Life Initiative.

**Abbreviations:** CHAI, Clinton Health Access Initiative; ED, emergency department; ETAT, Emergency Triage and Treatment; FGDs, Focus Group Discussions; ICA, inductive content analysis; IPD, inpatient department; LMICs, low- and middle-income countries; MOXY, Medical OXYgen improvement program; OPD, outpatient department; SRQR, Standards for Reporting of Qualitative Research; WHO, World Health Organization.

involved guards, lay bystanders, students, and caregivers. Duplication in assessment and treatment steps and misrouting from intended pathways occurred especially when initial care was spread across multiple locations. After-hours closure of OPDs shifted the place of entry to EDs or IPDs and could result in caregiver confusion. Administrative procedures (registration and payment) and recurrent stock-outs of medications and consumables could delay initiation of clinical processes. Referral pathways were inconsistent and some referrals were informal, undocumented, and enacted prior to patient stabilisation. Our findings are based on a purposive sample of facilities from Nigeria and Uganda, which may not be representative of other low-resource settings.

## Conclusions

Process mapping can help understand context and identify opportunities for intervention to improve facility care of severely ill children. We define organisational archetypes as heuristic tools for facility leaders and policymakers that can help facilities locate their configuration and recognise context-specific priorities. Potential low-cost opportunities for improvement include: building on existing adaptations (e.g., involving non-clinical staff and families in triage), formalising triage, streamlining non-clinical care processes that can delay clinical care (e.g., clearer signage and expedited administration), and strengthening referral systems.

## Author summary
### Why was this study done?

- Child mortality remains high in countries with weak emergency care systems.

- Half of these deaths occur in health facilities, most within 24 hours, and many are preventable with improvements in care.

- Research has focussed on single or packages of interventions. Little is known about how hospitals in low-resource settings are actually organised to provide emergency care for children.

### What did the researchers do and find?

- We studied 26 secondary and tertiary hospitals in Nigeria and Uganda.

- Embedded researchers mapped the steps that severely ill neonates (<29 days) and children (<15 years) took from arrival to admission, and documented staff, resources, and processes.

- We developed distinct 'archetypes' of facilities that can be used to understand the variability of where and how emergency care is delivered.

**What do these findings mean?**

- We describe four archetypes that can be used as a heuristic tool to help prioritise and contextualise interventions for emergency care.

- The findings highlight opportunities to identify and legitimise pre-existing informal adaptations, reorganise patient flow, prioritise clinical triage, and streamline non-clinical processes.

- Our findings are based on a purposive sample of facilities from Nigeria and Uganda, which may not be representative of other low-resource settings.

## Introduction

Under-five mortality in low- and middle-income countries (LMICs) remains 10–20 times higher than in high-income countries [1]. Half of these deaths could be prevented by improvements in clinical care within the first 48 hours [2,3]. Children at the highest risk of death are those with critical illness, which affects 30% of admissions to hospitals in low-resource settings [4].

Tackling this burden of preventable morbidity and mortality requires strengthening emergency response at all levels of the health system [5,6]. Hospital-based interventions, such as structured triage and treatment approaches [7–9] can improve care, but their implementation and effectiveness have been inconsistent across contexts [10]. Persistent gaps in facility readiness to provide time-critical care remain [11,12].

One constraint in designing and implementing effective interventions for emergency care has been limited understanding of the diversity of contexts in which interventions are introduced, including the organisational structures of health facilities [13]. Standardised tools such as the World Health Organization's (WHO) Emergency Triage and Treatment (ETAT) and related guidelines provide clinical direction to individual clinicians on the recognition and treatment of severely ill children [9]. However, they offer limited guidance for facilities seeking to improve how emergency care unfolds in their specific context. Facility readiness assessments similarly tend to focus on the presence of resources or adherence to protocols, rather than on how care is actually delivered in practice within complex health system environments.

In this study, we sought to describe recurring patterns of facility organisation and patient flow ('archetypes') to inform program theories for future interventions to improve emergency care for children.

## Methods

### Design

We used a qualitative, multi-method design informed by a critical realist ontology [14]. In this view, reality exists independent of the observer. Reality can be understood by theorising the mechanisms that interact with context to produce observable patterns. This study focussed on describing the empirical findings (the what, who and how) to support future explanatory realist enquiry (the why).

Our epistemology was pragmatic [15], as we aimed to make sense of patterns in facility organisation that would be useful for intervention design and system improvement.

Our data included facility process maps, produced by 16 researchers embedded in 26 facilities across 3 states in Nigeria (Kano, Rivers, Lagos) and 3 sub-regions in Uganda (Bugisu, Toro, Ankole). We collected data from 8th January 2024 to 6th December 2024 in Nigeria and 30th October 2023 to 6th December 2024 in Uganda. These included 23 secondary facilities (12 Uganda, 11 Nigeria) and 3 tertiary facilities (all in Nigeria), all with inpatient services for children (<15 years) (Table 1).

Table 1. Characteristics of participating facilities, sorted by country facility level, and patient load.

| | Facility level | Type | Number of beds* |
|---|---|---|---|
| **Uganda** | | | |
| HF1 | SS | Public | Paed 21; Neo 6 |
| HF2 | SS | Public | Paed 12; Neo 1 |
| HF3 | SS | Public | Paed 37; Neo 4 |
| HF4 | SS | Public | Paed 8; Neo 8 |
| HF5 | SS | Private | Paed 10; Neo 3 |
| HF6 | SS | Public | Paed 8; Neo 1 |
| HF7 | SS | Private | Paed 42; Neo 4 |
| HF8 | SS | Public | Paed 27; Neo 2 |
| HF9 | SS | Public | Paed 11; Neo 4 |
| HF10 | LS | Public | Paed 36; Neo 45 |
| HF11 | LS | Public | Paed 58; Neo 17 |
| HF12 | LS | Public | Paed 70; Neo 49 |
| **Nigeria** | | | |
| HF13 | SS | Public | Combined ward 17 |
| HF14 | SS | Public | 2 paediatric spaces in ED |
| HF15 | SS | Public | Paed 38; Neo 0 (all referred) |
| HF16 | LS | Public | Paed 15; ED 6; Neo 20 |
| HF17 | LS | Public | Paed 37; Neo 0 (all referred) |
| HF18 | SS | Public | Combined ward 16; Neo 0 (all referred) |
| HF19 | LS | Public | Paed 15; ED 14; Neo 20 |
| HF20 | LS | Public | Paed 19; ED 8; Neo 30 |
| HF21 | LS | Public | Paed 34; ED 21; Neo 15 |
| HF22 | LS | Public | Paed (ED) 32; Neo 35 |
| HF23 | T | Public | Paed 23; ED 8; Neo 24 |
| HF24 | SS | Public | Paed 11; ED 4; Neo 6 |
| HF25 | T | Public | Paed 25; Neo 37 |
| HF26 | T | Public | Paed 33; ED 9; Neo 17 |

*Excluding beds for inborn neonates outside of paediatric or emergency units (e.g., maternity or post-natal wards).

Abbreviations: LS, large secondary facility; SS, small secondary facility; T, tertiary facility. Paed, paediatric beds; Neo, neonatal beds; ED, emergency department.

We conducted Focus Group Discussions (FGDs) with embedded researchers on 10th February 2025 in Uganda and 1st March 2025 in Nigeria (Text C in S1 Appendix, p. 39). We used inductive content analysis (ICA) to analyse the data [16]. This study is reported as per the Standards for Reporting of Qualitative Research (SRQR) (S2 Appendix).

## Context

The study was conducted as part of the evaluation of a multi-country Medical OXYgen improvement program (MOXY) implemented by the Clinton Health Access Initiative (CHAI) in partnership with government. States and regions were identified by the MOXY program as high priority for intervention based on high clinical demand for oxygen or weak oxygen systems.

Our aim was not to select a nationally representative sample of hospitals in Nigeria or Uganda, but to understand diversity in facility contexts within the opportunity afforded by the program. Therefore, we included facilities of varying sizes and levels of resourcing. We categorised the facilities as small secondary (large health centres or general hospitals), large

secondary (provincial hospitals or general hospitals with specialist services) and tertiary hospitals (highest level of care, in urban centres). We purposively oversampled smaller, more remote facilities which are usually less represented in the emergency and critical care literature. Most included facilities were government-run; a small number of privately run facilities were included in contexts where these were the main paediatric inpatient care provider to the surrounding community.

## Theoretical approach

Emergency care for children is typically assessed and improved through the application of standardised tools such as WHO ETAT [9] or the Interagency Integrated Triage Tool [17]. These approaches posit that if children are triaged and managed using predefined signs and symptoms, and treated according to clinical algorithms, outcomes should improve. Performance is measured against normative standards, and deviations from guidelines are often seen as deficiencies. We expanded on this clinically-centric view, learning from experiences in implementing medical oxygen systems in LMICs [18–20]. We conceptualised emergency care not only as technical practices, but also as socially constructed processes. For example, whether and how critically ill patients receive care is not only shaped by healthcare worker knowledge and skills, or access to resources, but also by local norms, power dynamics, informal practices and adaptations, and caregiver preferences, perceptions and expectations.

## Researcher characteristics, reflexivity and positionality

Our team occupied a variety of insider and outsider positions [21], with diversity in origin (nationality, region, community ties), religion, social status (gender, age and ethnicity), professional role (clinical versus non-clinical, cadre and seniority) and stance (Text A in S1 Appendix, p. 28). We drew on Wilkinson's framework of reflexivity which distinguishes three forms: personal (identity, values), functional (research design and processes) and disciplinary reflexivity (traditions and norms of the discipline) [22]. Personal reflexivity: embedded researchers often lived in the same communities as participants, accessed the same health services for their own families, and held pre-existing attitudes towards facilities and staff. This positioning enabled trust, tacit understanding, and access, but also meant that their interpretations were shaped by community perspectives and personal relationships. Research supervisors (both internal and external to country) approached the study with personal and professional investments in health systems research, and their interpretations were shaped by earlier conceptualisations of what the problems in health facilities were and how solutions should be framed. Functional reflexivity: the organisation of the study was influenced by pragmatic decisions about where to conduct research. Site selection was based on where the MOXY program was operating, alongside considerations of security and accessibility. Disciplinary reflexivity: because research methods training was delivered by clinicians to research assistants with clinical backgrounds (Table I in S1 Appendix, p. 25), events were often viewed through a clinical lens, with emphasis on patient care processes, risks, and clinical priorities.

## Data collection

We used the process mapping method [23], and adapted the process mapping framework described by Antonacci and colleagues [24]. We set out four phases to guide embedded researchers and conducted regular training (Text B in S1 Appendix, p. 29).

Embedded researchers worked in teams of two to three, with each team assigned to a cluster of facilities—four per team in Uganda and five per team in Nigeria. They completed two 1-month rotations at every assigned site (Fig 1). We developed and used a standardised, semi-structured data collection tool (Text B in S1 Appendix, p. 29). We encouraged the facility-embedded researchers to remain open to recording observations beyond the scope of the tool.

We conducted FGDs with facility-embedded researchers at the project's endpoint to gain an in-depth and contextual understanding of the process maps. FGDs followed a semi-structured discussion guide (Text C in S1 Appendix, p. 39). Discussions were held in English and were facilitated by four members of the research team (local and international)

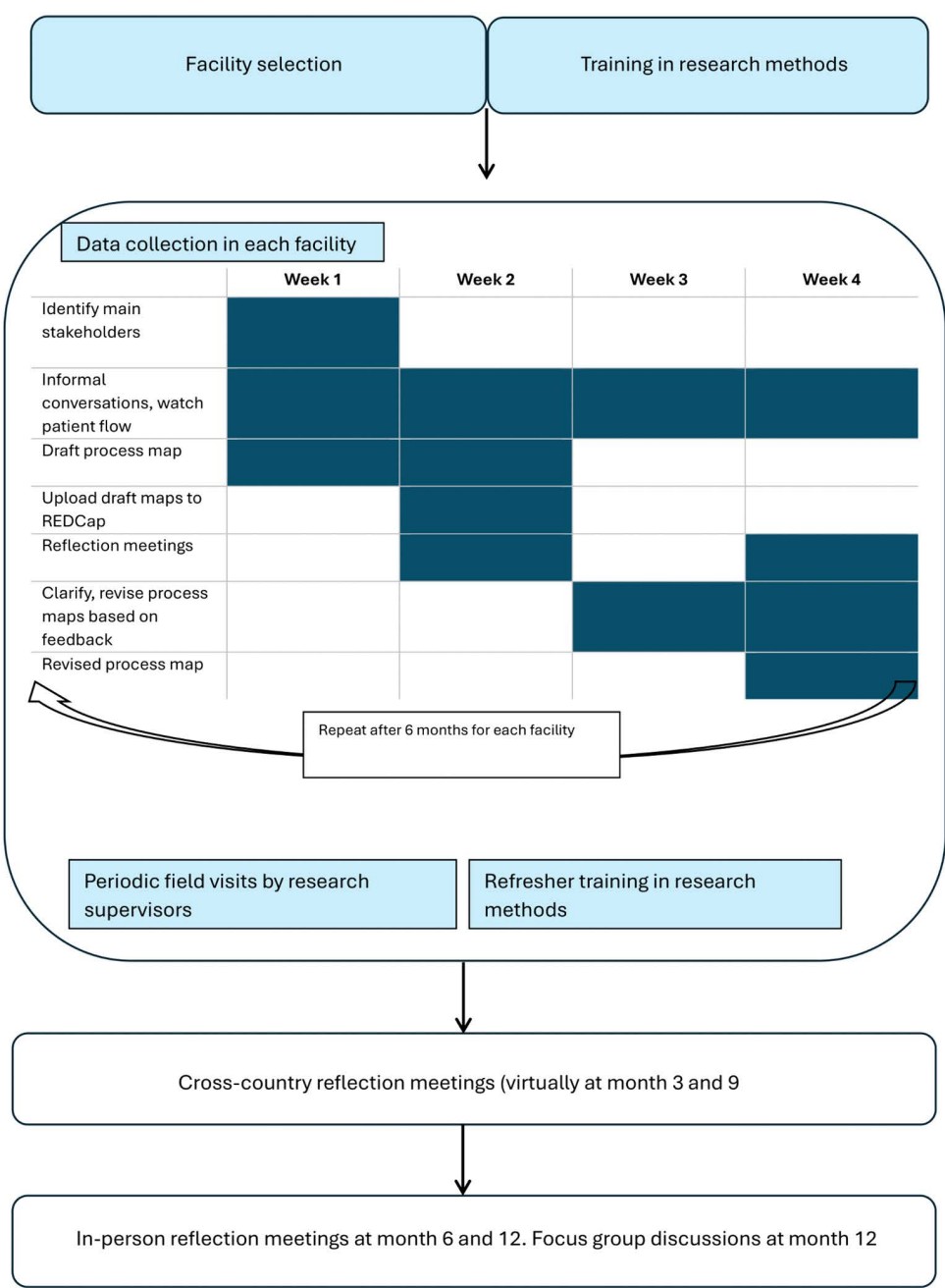

**Fig 1. Research outline.** Following facility selection, embedded researchers were assigned to a cluster of facilities (four per team in Uganda, and five per team in Nigeria). Training in research methods took place prior to data collection, and then periodically refreshed. Researchers were embedded within facilities for 4-week periods, and rotated across their assigned facilities, completing 2 rotations at each assigned site. Oversight by each country's research supervisors was via virtual meetings, regular phone calls and text messages and field visits. Virtual cross-country seminars were held at month 3 and 9 of data collection. In-country meetings by Nigeria/Uganda and international research team took place in month 6 (with research refresher training) and month 12. Focus Group Discussions with embedded researchers were conducted in month 12.

experienced in qualitative methods (FS, DM, RS and MB in Uganda; and AAB, AS, RS, MB in Nigeria). FGDs lasted approximately 60–90 min. Written informed consent was obtained from all participants prior to participation (Text D in S1 Appendix, p. 41). With permission, sessions were audio-recorded, transcribed verbatim, and anonymised.

## Data processing and analysis

Hand-written/drawn process maps were uploaded to a secure data management platform (REDCap, Research Electronic Data Capture) and transcribed for analysis (Fig A in S1 Appendix, p. 26–27). These, along with anonymised transcripts of the FGDs, were uploaded to NVivo for analysis.

We adopted ICA because our aim was to describe facility processes and organisations to practically guide intervention design, rather than to develop high-level theory [16]. Moreover, we wanted to be led by the data, rather than pre-formed ideas of organisational patterns of facilities. Data were coded independently by RS, AS, DM and MB.

We read and re-read the process map for each facility individually, and applied deductive codes (Table A in S1 Appendix, p. 2), to organise the data according to discrete steps in the patient journey. We then coded inductively to develop categories such as "location of triage". We iteratively recoded the text to develop sub-categories (e.g., "triage in OPD"), then compared these sub-categories across facilities to find groupings or archetypes (e.g., "OPD screen and treat"; Table 2 as an example). We represented these archetypes graphically using Swimlane diagrams to show the flow of personnel, locations and activities.

FGDs were coded in a similar way, but without the initial step of deductive coding. We included descriptions of models for neonatal care. As we did not set out to study obstetric and maternal services in each facility, the scope of neonatal data was more limited. We described neonatal 'pathways' specifically focussed on patient flow and location of care.

Quotations are the words of the facility-embedded researchers, taken from their narrative descriptions of the facilities in process mapping or the FGD. In reporting illustrative quotes, we distinguished between excerpts from process-map narratives (indicated by "facility code and country") and those from FGDs (indicated by "FGD, facility code and country").

## Ethical approval

The research was approved by the relevant ethics boards in each country—University of Ibadan/University College Hospital Nigeria (ref: UI/EC/23/0535) and Makerere University School of Health Sciences Uganda (ref: MAKSHSREC: 2023-598)—and by the Human Research Ethics Committee, Royal Children's Hospital Melbourne, Australia (ref: HREC/97817/

**Table 2. Example stages of analysis used to generate organisational archetypes (Archetype B).**

| Stage of analysis | Description | Example |
|---|---|---|
| Deductive coding | Process maps organised using pre-specified steps in the patient journey through a facility (e.g., entry, triage, initial treatment, admission) to enable comparison across facilities | Codes for "facility entry", "initial assessment", "treatment location" |
| Inductive coding | Inductive coding applied to describe how care was organised within each step | Codes for location of care ("assessment in OPD"), personnel delivering it ("clinical officer assessment"), and how that care is actually delivered ("formal triage step, measuring vital signs, allocating triage category") |
| Patterns of organisation | Codes grouped into higher-order categories describing patterns of organisation | "Separate assessment and treatment locations" |
| Archetype generation | Higher-order categories compared across facilities to synthesise recurring archetypes representing typical models of care | Archetype B: "OPD screen and send" (assessment in OPD, treatment in IPD) |

OPD, outpatient department; IPD, inpatient department.

RCHM-2023) and the Swedish Etikprövningsmyndigheten (ref: 2024-00868-01). Written informed consent was sought from facility leadership prior to participation (Table D in S1 Appendix, p. 41).

## Results

### Facility organisation for emergency care

**Facility archetypes for children (≥29 days to 14 years).** Archetype A ('OPD screen and treat') describes facilities where all patient—including critically ill children—were assessed and managed in Outpatient Departments (OPDs) (Fig 2 and Table B in S1 Appendix, p. 3). This model was found in three small secondary facilities in Uganda (HF3, HF5 and HF9). Clinical officers—mid-level clinicians with a three-year diploma in clinical medicine and community health [25]—played a central role in developing management plans, providing emergency care, and determining patient disposition. Doctors were usually based in inpatient departments (IPDs) and called on when needed. During the study period, one facility (HF9) decommissioned its emergency space due to staffing constraints, shifting emergency services to IPDs.

Archetype B ('OPD screen and send') refers to facilities where assessment and planning for emergency treatment occurred in OPDs, followed by transfer to IPDs for management (Fig 3 and Table C in S1 Appendix, p. 4). This was common in small secondary facilities in Uganda. In both archetype A and B facilities, after-hours patients were expected to present directly to IPDs.

Archetype C ('ED receive and treat') represents facilities with dedicated Emergency Departments (ED) providing emergency assessment and management (Fig 4 and Table D in S1 Appendix, p. 7), with separate outpatient spaces for ambulatory care. This model characterised larger secondary and tertiary facilities in Nigeria. Some archetype C facilities also ran paediatric OPDs, which referred ill children to EDs and received children suitable for ambulatory care from ED.

Archetype D ('IPD receive and treat') describes facilities where IPDs provide both assessment and emergency management (Fig 5 and Table E in S1 Appendix, p. 10). We identified this model in two large secondary facilities in Uganda, each with paediatric wards designed with dedicated triage and emergency spaces. In one facility (HF10), the paediatric IPD received and managed all patients, offering both ambulatory and emergency care. The second facility (HF12) operated a separate OPD for ambulatory cases, located away from the IPD.

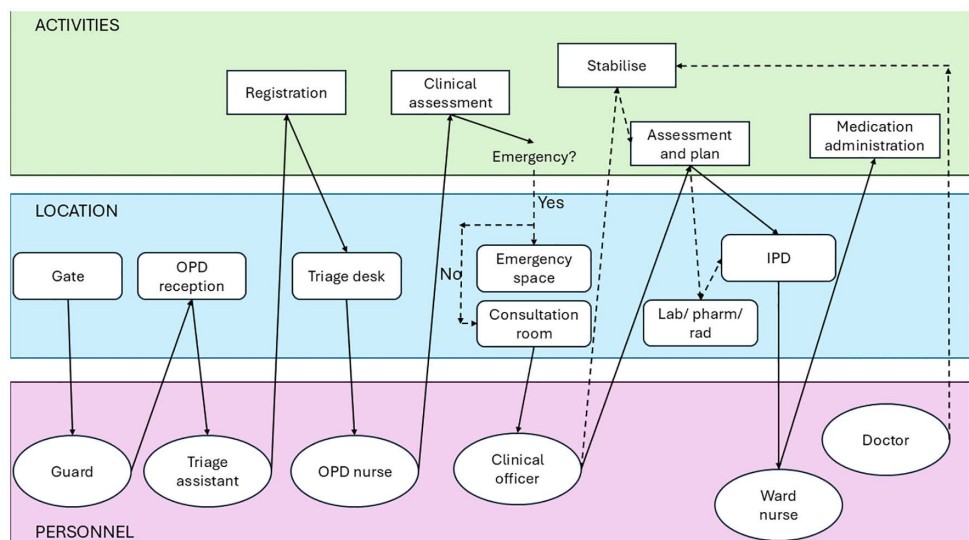

**Fig 2. Swimlane diagram for facilities aligning with Archetype A ('OPD screen and treat').** Row colours correspond to activities (green), location (blue) and personnel (purple). OPD: outpatient department; IPD: inpatient department; Lab/Pharm/Rad: laboratory, pharmacy and radiology.

**Fig 3. Swimlane diagram for facilities aligning with Archetype B ('OPD screen and send').** Row colours correspond to activities (green), location (blue) and personnel (purple). OPD: outpatient department; IPD: inpatient department; Lab/Pharm/Rad: laboratory, pharmacy and radiology.

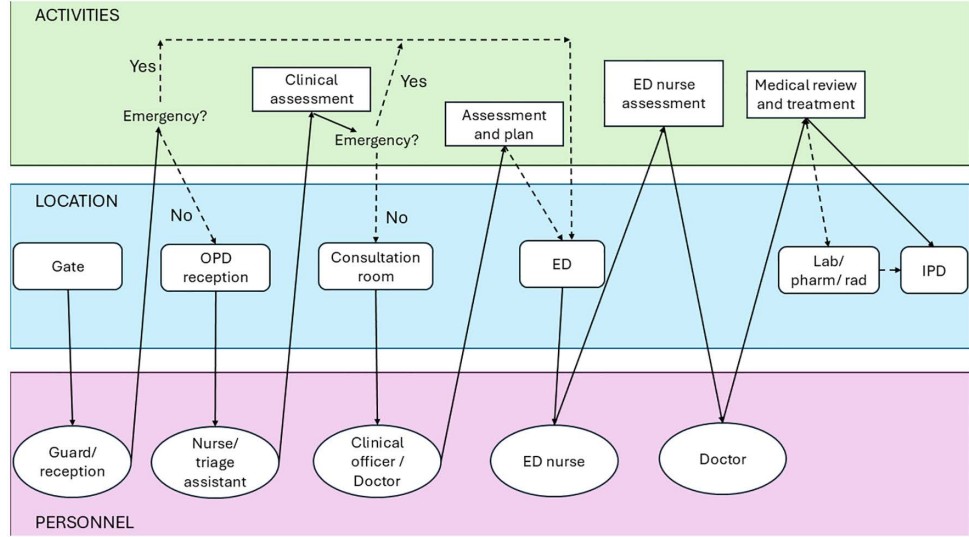

**Fig 4. Swimlane diagram for facilities aligning with Archetype C ('ED receive and treat').** Row colours correspond to activities (green), location (blue) and personnel (purple). ED: emergency department; OPD: outpatient department; IPD: inpatient department; Lab/Pharm/Rad: laboratory, pharmacy and radiology.

Archetypes reflect intended models of care, but practices sometimes varied. For instance, in archetype A facilities, caregivers may bypass OPDs and present directly to IPDs. Special populations, including children with trauma or surgical conditions, had different models of care, often sharing services with adults (Table G in S1 Appendix, p. 16).

**Neonatal pathways (<29 days).** Facilities managed neonates in four main ways: (a) in dedicated neonatal units (NNUs); (b) in maternity units with midwife-led care; (c) in EDs for neonates presenting from the community ('outborns')

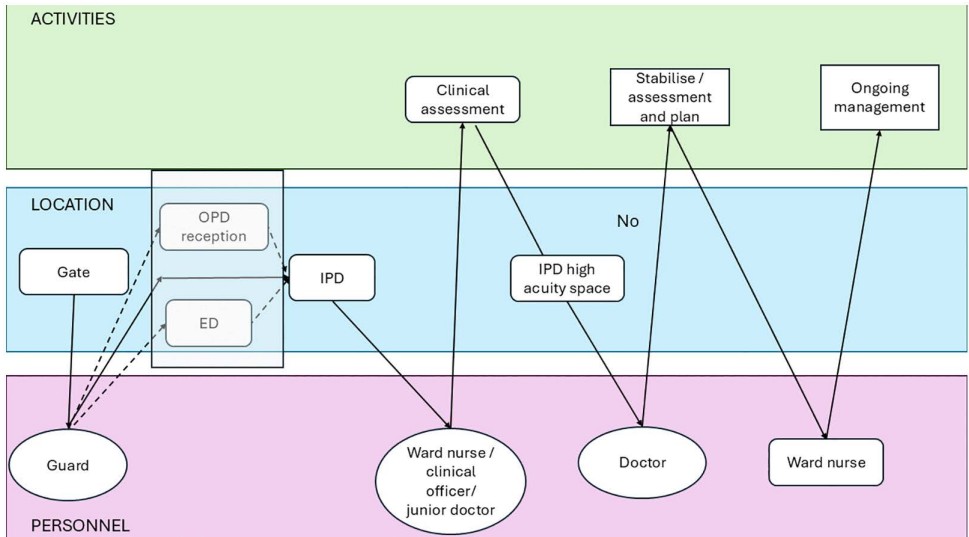

**Fig 5. Swimlane diagram for facilities aligning with Archetype D ('IPD receive and treat').** Row colours correspond to activities (green), location (blue) and personnel (purple). ED, emergency department; OPD, outpatient department; IPD, inpatient department; Lab/Pharm/Rad, laboratory, pharmacy and radiology. The indented pathway is from gate to IPD. Other possible pathways include children presenting to OPD and ED (within shaded grey box), not fully represented in the diagram.

and IPDs for those born in the facility ('inborns'); or (d) referring all ill neonates elsewhere (Table F in S1 Appendix, p. 11). In most small secondary facilities in Uganda, sick neonates (inborns and outborns) were cared for by midwives in maternity units. In Nigeria, EDs typically took on the care (or at least the initial assessment) of ill outborns. The three facilities that referred all neonates were located in Nigeria (HF14, HF15 and HF17).

**Processes throughout the patient journey**

**Facility entry and navigation—"They do not know where to present".** Facility entry and navigation was influenced by caregiver knowledge of the facility from prior encounters, and opportunistic advice from staff and bystanders. In most facilities, the security guard was the first point of contact. Navigation challenges were common. In one extreme example from a large tertiary facility:

*"For sick children and their caregivers walking through the facility gate, and they hypothetically know their way around the facility, it will take them about 30-40 minutes or more to walk to the children's OPD [outpatient department] and if they don't know the facility, maybe over an hour."* FGD, referencing HF25, tertiary facility, Nigeria.

We observed differences after-hours, when OPDs were closed and patients were implicitly expected to present directly to EDs or IPDs. This was especially relevant to small secondary facilities, and could result in significant delays and caregiver distress (Table H in S1 Appendix, p. 18).

Perceived acuity by guards, bystanders and/or clinical staff—even prior to initial assessment—often led to the child being directed to ED or IPD, irrespective of the intended pathway of care (Table H in S1 Appendix, p. 18). The location of first presentation was typically guided by the caregiver's prior knowledge of the facility.

We observed major changes in facility structures in 5 facilities during the study period, triggered by acute staff shortages (HF9, HF13, HF18), equipment breakdown in emergency spaces (HF9) or renovation (HF14) and expansion (HF16). In one example, this led to closure of a ward, and suspension of all paediatric admission services (HF14).

**Triage—"It's like an understanding".** Triage—sorting patients according to acuity of presentation using a structured approach—occurred multiple times in different forms across different areas. This could occur at the facility entrance, waiting room, OPD or ED. Multiple personnel could be involved: security guards, bystanders, community health workers, students, nurses and doctors (Table H in S1 Appendix, p. 20).

In practice, even when adopted in theory, no facility was consistently using formal triage systems. Barriers to implementation of triage systems included workload and unpredictable staffing, reliance on untrained rotating clinical staff and non-clinical staff, and concerns that some caregivers would resist an approach that sees some children seemingly skip a line (Table H in S1 Appendix, p. 19).

The first opportunity for triage was usually at the point of facility entry, with pre-triage assessments by security guards and information desk assistants. Clinical assessments ranged from informal general inspections to structured measurement and documentation of vital signs, and could occur across different spaces and providers, especially for archetype B and D facilities. This risked duplication and delays in management. Healthcare workers, especially when overwhelmed by patient load, would usually revert to a first-come first-served approach (Table H in S1 Appendix, p. 20).

To counteract navigation challenges for caregivers, facilities employed informal strategies to help identify and correctly stream the unwell patient. One strategy involved periodic waiting room announcements by staff inviting caregivers to voice concerns. These practices were usually tacit and unwritten:

*"It's like an understanding … even as the doctors are working and the nurses, you can hear someone say, someone, there's no one on triage, someone check … the bench. So it's something that is known by the health workers, although it's not written on the wall or something."* FGD, referencing multiple facilities including HF4 (small secondary facility) and HF12 (large secondary facility), Uganda.

Caregiver distress, convulsions, a child's inability to stand and walk, and high fever were reported by health facility staff as important emergency signs (Table H in S1 Appendix, p. 20). Waiting spaces that mixed scheduled outpatient visits with acute presentations made oversight more challenging. As a result, healthcare workers were often conflicted between running busy OPDs, and attending to severely ill arrivals.

In the context of staff shortages, smaller facilities reserved their most experienced staff to IPDs, leaving less trained staff to receive patients and conduct initial assessment (Table H in S1 Appendix, p. 22). These cadres often performed outside their scope of practice (Table H in S1 Appendix, p. 21).

**From triage to stabilisation—From here to there and back again.** We observed the important role of non-clinical staff in the timeliness of emergency care, including administration, payment and logistics.

Administrative steps were often required before diagnostics or treatments, delaying emergency care (Table H in S1 Appendix, p. 23). These steps included registration and consultation payments. Other requirements of caregivers, such as the need to purchase exercise books for clinical documentation, could further delay clinical care. At times, these delays were further exacerbated by inaccurate or confusing advice provided to caregivers by facility staff or bystanders.

Conversely, timely emergency care was facilitated in some facilities by co-locating payment and administrative desks within clinical areas and conducting registration in parallel with triage and emergency care (Table H in S1 Appendix, p. 23). In one facility, children with emergency conditions were given a red card to signify the need to expedite administration and payment for registration, investigations and medications (HF16, Nigeria).

Children presenting to archetype B and C facilities were at risk of delayed emergency management for different reasons: due to delayed recognition of critical illness in OPD in B facilities (delay in streaming), and due to caregivers incorrectly presenting to OPD (rather than ED) in C facilities. These problems were exacerbated by poor communication and limited handover between staff across clinical areas, as well as competing priorities of inpatient staff—particularly in archetype B and D facilities. More established communication processes were observed in larger secondary and tertiary facilities, which tasked students or support staff to channel information between wards.

***Access to emergency therapies—"Left for the patient to decide where they will purchase the drug".*** Emergency medicines were commonly out of stock or not restocked, and sourcing them became the responsibility of caregivers, often from external pharmacies (Table H in S1 Appendix, p. 22). Caregivers could be expected to replace used emergency medicines. This often did not occur due to financial constraints. Healthcare workers also reported a reluctance to ask the family of a deceased child to payback emergency medications.

The process for obtaining emergency medicines can be complex, involve multiple steps, and often relied heavily on the caregiver:

*"If drugs are prescribed for the patient, it's left for the patient to decide where they will purchase the drug, either at the facility pharmacy or outside the pharmacy but on observation, most times, the pharmacy doesn't have the prescribed drugs and the patients are left with no other option but to go outside the facility to get it."* HF13, Nigeria.

***Emergency management and scope of practice—Going above and beyond.*** Healthcare workers often functioned beyond the scope of their formal role in providing emergency management (Table H in S1 Appendix, p. 21), sometimes with limited supervision:

*"Whenever a doctor is needed to review a patient at the Neonatal Intensive Care, the nurses on duty or midwife on duty makes a phone call to any of the doctors on duty attending to other wards… however most of the time, sick neonates are only managed by the nurses and midwives."* HF1, Uganda

***Investigations—Caregiver burden.*** Investigations, especially pathology, often relied on the caregiver: to carry the sample to pathology, await and chase results, and relay the results back to clinical staff. In most Nigerian facilities, there were additional steps of costing and making payment for the test (Table H in S1 Appendix, p. 23).

***Referral—Helpful but difficult to do safely.*** Two types of referrals were observed: a formal referral following clinical assessment with a letter outlining reasons for referral, which may include access to surgical or subspecialty services, referral for oxygen or referral because the facility was at full capacity. There were also 'informal' and usually undocumented referrals. Caregivers would be told to present elsewhere, often prior to clinical assessment and stabilisation. When this occurred, it was due to lack of bed capacity, lack of services to manage certain populations (e.g., neonates) and lack of doctors after-hours (Table H in S1 Appendix, p. 23). Doctors usually needed to approve formal referrals from larger facilities, and this could lead to significant delays if doctors were unavailable.

Ambulance transfers involved direct and indirect costs (e.g., payment for fuel or payment to wash the ambulance). Transfers rarely had a clinical escort, often because of staff shortages. One facility described having to make the choice between leaving the inpatient ward unstaffed or transferring patients without a clinical escort (HF8, Uganda). Private transfers arranged by caregivers commonly took place and were usually poorly documented.

## Discussion

This multi-method qualitative study provides a detailed description of how secondary and tertiary health facilities in Nigeria and Uganda are organised to deliver emergency care for neonates and children. Our findings can help clinicians, facility leaders, policymakers and implementers conceptualise the varied ways emergency care systems operate. The proposed facility archetypes should be understood as heuristic tools to support process understanding, rather than as fixed or country-specific classifications. The same facility may shift between archetypes over time, or in response to day-to-day variation or specific circumstances.

Our analysis highlights four lessons. Firstly, process mapping can help facilities identify and prioritise context-specific areas for improvement. By locating their organisation within the proposed archetypes, facility leaders can focus on a

smaller set of interventions most likely to have impact. For instance, facilities where initial assessment and treatment occur in separate locations (e.g., archetype B) face delays in emergency management and should prioritise reconfiguring patient flow. Likewise, in facilities where clinicians in busy OPDs or IPDs juggle competing responsibilities, emergency care is easily deprioritised or delayed, highlighting the need to protect dedicated emergency care resources. Experience from Kenya has shown how ownership and buy-in of emergency care interventions can be compromised in settings like mixed OPDs if staff do not perceive the care of critically sick children as their core business [26].

Secondly, we observed that facilities develop adaptations that address local challenges, including task shifting and delegation. Where these adaptations are beneficial, they risk being undermined by interventions that are designed and implemented without effective co-design or understanding of context [27–29].

Task-shifting has been extensively studied. It is often adopted informally as a pragmatic response to workforce shortages [30,31]. Published experience with lay-personnel triage [32,33], nurse-led emergency assessment and management [34], vital sign monitoring and response systems by volunteers [8,35], and family-assisted patient monitoring [36] have demonstrated promising results. But not all applications of task shifting are appropriate or desirable. Concerns have been raised regarding compromised quality and safety of care to patients [37], as well as risk of burn-out among those to whom tasks are shifted [38].

Our observations indicate that informal task-shifting is already occurring in day-to-day practice. However, this reality is rarely reflected in formal policies and lacks the legitimacy afforded by institutional recognition, regulatory frameworks, and leadership endorsement [30,39]. The lack of visibility limits the ability of health systems to define role boundaries, ensure accountability, and determine which clinical functions can be safely shifted or shared.

Practically, our findings suggest that quality improvement efforts in emergency care should begin with a diagnostic phase that examines work as actually done rather than work as intended. Guidelines and mandates should acknowledge the gap between what is recommended and the capacity of those available to deliver it. Our findings suggest a need to consider whether official scopes of practice align with the realities of everyday clinical work, while maintaining clear standards of quality of care and patient safety.

Thirdly, our observations have highlighted that triage was rarely a formalised event, with consistent absence of standardised triage systems. Instead, patient assessments occurred in fragmented ways, shaped by a wide range of individuals including caregivers and non-clinical staff. This reinforces the notion that triage is a shared responsibility [40]. However, beyond training, which has often been developed from a clinical lens for clinicians [9], there is still little evidence to guide facilities on how to operationalise this principle effectively.

Fourthly, because emergency care unfolds across multiple locations [41], and can vary by time of day and patient population, interventions confined to a physical location will usually not be sufficient. Moreover, seemingly minor non-clinical aspects like facility signage or clerical registration can be as important as clinical capability in providing timely care. These non-clinical aspects have largely been considered only peripherally in quality improvement programs [32,42,43]. In themselves, they offer opportunities for low-cost and potentially effective interventions.

We observed weak referral systems, including informal referrals without documentation, handover or pre-referral stabilisation, consistent with published literature [44,45]. This population is not captured in routine health information systems, and contributes to the missed burden of preventable deaths. The requirement for medical authorisation to refer patients out of small facilities, even when physicians were not reliably available, likely contributes to additional delays. Access to an equipped ambulance service was frequently limited. These gaps echo WHO Emergency Care System Framework [46], which identifies coordinated referral and transport as core system functions.

While previous quantitative data have assessed facility readiness assessments for emergency care [47,48], our data capture the dynamic aspects of care organisation. Researchers were embedded within each facility over extended periods, gathered the perspectives of multiple cadres of staff and caregivers, observed care in real time, and iteratively refined the process maps in consultation with facility teams. Regular reflection meetings with the research team facilitated

cross-site comparison and helped build a nuanced description of the organisational and contextual factors involved in emergency care delivery.

There are a number of considerations and limitations. Our understanding of processes is partly constructed from data observed and reported by embedded researchers. This is shaped by their positionality as researchers, clinicians and members of the studied community, and influenced by the views of supervisors and lead researchers. Embedded researchers had access to information on care as intended (what was reported by healthcare workers or documented in policies), and care as done (what they observed). The documented observations may conflate these two concepts. We did not conduct in-depth interviews with facility staff to understand decision-making, motivations, and individual barriers and enablers. As neonatal care pathways are closely intertwined with midwifery and maternity services, and were not the primary analytic focus, our descriptions of neonatal care organisation are less detailed.

Process mapping provides a practical way to understand how facilities organise care for severely ill children. By identifying archetypes of facility organisation, we highlight context-specific priorities for improvement. These include building on existing adaptations, formalising triage, streamlining non-clinical care processes, and strengthening referral systems. The findings have potential utility for national emergency care planning and hospital accreditation processes, particularly in conceptualising readiness beyond the presence of resources alone.

### Research outline

*Three facilities align with Archetype A for children (≥29 days), all small secondary facilities in Uganda: HF3 and HF9 (government-run) and HF5 (private not-for-profit).* In these facilities, outpatient departments (OPD) manage all comers, with a dedicated emergency space where the sickest are assessed and managed. Nurses/nurse assistants (± lay-personnel) 'triage' patients, an activity which involves a combination of patient registration, general assessment for obviously unwell presentation (e.g., convulsions or coma) and vital signs. At the point of nurse assessment, patients identified as being critically unwell are sent directly to the emergency space for review by clinical officers. Doctors are primarily responsible for inpatient departments (IPD) and called on as required for additional assistance with assessments or resuscitation. After-hours, care shifts to IPDs, except for HF5, which runs a 24-hour OPD.

During the study period, the emergency room of HF9 became non-functional. Patients needing urgent assessment and treatment sent directly to IPD.

*Eight facilities align with Archetype B for children (≥29 days), all small secondary: 6 in Uganda (HF1, HF2, HF4, HF6, HF7, HF8), and 2 in Nigeria (HF13 and HF18).* In these facilities, outpatient departments (OPD) serve dual functions of providing ambulatory care and identifying and sending severe presentations to a different space (usually IPD) for further assessment and management. After-hours care shifts entirely to inpatient ward (or separate Emergency Department [ED] as for HF13).

In Uganda, initial assessment by nurses, community extension officers or nursing students in OPD during day-time hours (up to 5 or 6 pm). Further clinical assessment by clinical officers, with doctors available onsite nearby or on-call. Few facilities have stabilisation or procedure rooms in OPD (HF6 and H7), but for all, treatment is provided in inpatient units or adult emergency departments (for older children).

In Nigeria, nurses triage, and doctors develop assessments and plans. In HF13, an OPD for children runs in parallel with a mixed ED, and so severely unwell children identified initially in OPD are streamed to ED for further management (in contrast to Archetype C facilities, where the intended pathway is initial presentation to ED).

In Archetype B facilities, if patients are identified as critically unwell early (e.g., at the gate by lay personnel), OPD can be bypassed, leading to reduced delays in reaching the location capable of providing emergency management.

*Thirteen facilities align with Archetype C for children (≥29 days). One in Uganda (HF11, large secondary facility) and 12 in Nigeria: 3 small secondary (HF14, HF15 and HF24), 6 large secondary (HF16, HF17, HF19, HF20, HF21, HF22), and 3 tertiary facilities (HF23, HF25 and HF26).* In these facilities, ED functions as the central

unit for emergency assessment and treatment. In a sub-group of facilities, there is a close interplay with OPD running concurrently which functions to stream severely unwell patients presenting to OPD. Similarly, ED can refer patients suitable for ambulatory care to OPD.

Models of care including shared adult/paediatric ED (e.g., HF14), paediatric ED within a larger adult ED (e.g., HF15) or a stand-alone paediatric ED (e.g., HF19). In Nigeria, EDs can take on short-stay admission for children (e.g., HF15, HF19 and HF22).

***Two facilities align with Archetype D for children (≥29 days). Both are large secondary facilities in Uganda (HF10 and HF12).*** In these facilities, the paediatric IPD manages the triage, initial assessment and management of severely unwell children. In HF10, an 'OPD' collocated within the ward functions as both a triage desk, and a consultation space for outpatients. The emergency treatment space is a high acuity space adjacent to the triage desk, within the paediatric IPD. In HF12, OPD is a separate space, and the paediatric IPD is configured to include an 'admission room' for triage and emergency management. In both facilities, resourcing for emergency and inpatient care are shared. Both facilities also run adult emergency departments (EDs), which care for adolescents, children with road traffic accidents, and often, children with surgical conditions. Children with medical conditions presenting to ED are redirected to the IPD.

## Supporting information

**S1 Appendix. Table A.** Deductive codes, following steps of the patient journey through a health facility. **Table B.** Description of facilities that align with Archetype A. **Table C.** Description of facilities that align with Archetype B. **Table D.** Description of facilities that align with Archetype C. **Table E.** Description of facilities that align with Archetype D. **Table F.** Initial assessment, treatment and admission locations for neonates (<29 days). **Table G.** Management of 'special' populations: older children, trauma and surgical conditions. **Table H.** Quotes from focus group discussions with researchers, organised under categories that explain steps in the patient journey through a facility. **Table I.** Characteristics of researchers participating in focus group discussions in Uganda and Nigeria. **Figure A.** Example of the evolution of process maps over time, from simple spatial maps of facility organisation **(A)** to conceptual maps of processes **(B)**. Maps were supplemented by narrative descriptions. **Text A.** Researcher positionality. **Text B.** Process mapping data collection tool. **Text C.** Focus group discussion guide. **Text D.** Consent procedures and resources.
(DOCX)

**S2 Appendix. Standards for Reporting Qualitative Research (SRQR) checklist.**
(DOCX)

## Acknowledgments

We thank the staff and leadership of participating health facilities in Nigeria and Uganda, and the Ministries of Health in both countries, for their support of this study. We also acknowledge the support of the Clinton Health Access Initiative teams in Nigeria, Uganda, and globally. We are grateful to Trevor Duke for his feedback on the manuscript.

The MOXY Implementation Research Collaboration includes

Embedded researchers in Nigeria: Agnes Adenike Amos, Muhammad Hassan Nafiu, Olamide Eunice Ajayi, Tina Ada Anyanwu, Elizabeth Emuze, Margaret Bassey Omoboriowo, Rosemary Nkemelu Samuel, Joseph Kereem Abuo, Agnes Ene Otobo, Temitope Motunrayo Idowu

Embedded researchers in Uganda: Khabitsa Bernard, Susan Mary Elizabeth, Marion Birungi, Namuyingo Ann, Mukundwa Awino Yvette, Mukhaye Mary, Priscilla Allipo, Marvice Namande, Anna Maria Namuyingo

Oxygen for Life Initiative: Ayobami A Bakare, Adegoke G Falade, Abiodun Sogbesan,

Makerere University: Freddy E Kitutu, Freddie Ssengooba, Dan Muramuzi, Elizabeth Ayebare,

Melbourne Children's Global Health: Mikael Burhin, Hamish Graham, Rami Subhi, Priya Kevat, Belinda Dawson-McClaren, Sumit Kane

Clinton Health Access Initiative: Eva Drucker, Felix Lam, Harriet Webster

Karolinska Institutet: Carina King, Sibylle Herzig van Wees,

CHAI & MOH Nigeria: Chizoba Fashanu, Maxwell Onuoha, Lekia Nwidae (CHAI Nigeria); Gilbert Shetak (Federal Ministry of Health and Social Welfare, Nigeria)

CHAI & MOH Uganda: Yewande Kamuntu, Blasio Kunihira, Santa Engol (CHAI Uganda); Charles Olaro (Uganda MOH)

## Author contributions

**Conceptualization:** Rami Subhi, Abiodun Sogbesan, Dan Muramuzi, Mikael Burhin, Ayobami A. Bakare, Adegoke G. Falade, Freddy E. Kitutu, Freddie Ssengooba, Carina King, Sumit Kane, Belinda Dawson-McClaren, Hamish R. Graham.

**Data curation:** Rami Subhi, Abiodun Sogbesan, Dan Muramuzi, Mikael Burhin, Ayobami A. Bakare.

**Formal analysis:** Rami Subhi, Abiodun Sogbesan, Dan Muramuzi, Mikael Burhin, Ayobami A. Bakare, Carina King, Hamish R. Graham.

**Funding acquisition:** Rami Subhi, Mikael Burhin, Ayobami A. Bakare, Adegoke G. Falade, Freddy E. Kitutu, Freddie Ssengooba, Carina King, Hamish R. Graham.

**Investigation:** Rami Subhi, Abiodun Sogbesan, Dan Muramuzi, Mikael Burhin, Adegoke G. Falade, Freddy E. Kitutu, Freddie Ssengooba, Carina King, Hamish R. Graham.

**Methodology:** Rami Subhi, Carina King, Sumit Kane, Hamish R. Graham.

**Project administration:** Rami Subhi, Abiodun Sogbesan, Dan Muramuzi, Mikael Burhin, Ayobami A. Bakare, Adegoke G. Falade, Freddy E. Kitutu, Freddie Ssengooba, Hamish R. Graham.

**Resources:** Hamish R. Graham.

**Supervision:** Rami Subhi, Abiodun Sogbesan, Dan Muramuzi, Mikael Burhin, Ayobami A. Bakare, Adegoke G. Falade, Freddy E. Kitutu, Freddie Ssengooba, Sumit Kane, Belinda Dawson-McClaren, Hamish R. Graham.

**Validation:** Rami Subhi, Hamish R. Graham.

**Visualization:** Rami Subhi, Carina King, Belinda Dawson-McClaren.

**Writing – original draft:** Rami Subhi.

**Writing – review & editing:** Rami Subhi, Abiodun Sogbesan, Dan Muramuzi, Mikael Burhin, Ayobami A. Bakare, Adegoke G. Falade, Freddy E. Kitutu, Freddie Ssengooba, Carina King, Sumit Kane, Belinda Dawson-McClaren, Hamish R. Graham.

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
