## [Editor Report · Decision Letter 0]

13 Nov 2025

Dear Dr Subhi,

Thank you for submitting your manuscript entitled "Pathways of emergency care for severely ill children in Nigerian and Ugandan hospitals: a process mapping study." for consideration by PLOS Medicine.

Your manuscript has now been evaluated by the PLOS Medicine editorial staff, and I am writing to let you know that we would like to send your submission out for external peer review.

However, before we can send your manuscript to reviewers, we need you to complete your submission by providing the metadata that is required for full assessment. To this end, please login to Editorial Manager where you will find the paper in the ‘Submissions Needing Revisions’ folder on your homepage. Please click ‘Revise Submission’ from the Action Links and complete all additional questions in the submission questionnaire.

For clinical studies, please upload a copy of your trial study protocol as a supporting information file. The study protocol should be the version submitted for approval to the institutional review board or ethics committee, should include any amendments to the study protocol, as well as the date of their approval by the institutional review or ethics committee. Please also detail any deviations from the study protocol in the Methods section of your manuscript. The editors will consider the protocol and study conduct prior to a final decision for external review.

Please re-submit your manuscript within two working days, i.e. by Nov 17 2025 11:59PM.

Kind regards,

Helen Howard

PLOS Medicine

---

## [Decision Letter · Decision Letter 1]

3 Feb 2026

Dear Dr Subhi,

Many thanks for submitting your manuscript "Pathways of emergency care for severely ill children in Nigerian and Ugandan hospitals: a process mapping study." (PMEDICINE-D-25-03978R1) to PLOS Medicine. The paper has been reviewed by subject experts and a statistician; their comments are included below and can also be accessed here: [LINK]

As you will see, the reviewers commended your work and provided very positive feedback on the methodology and clarity of the paper. Nonetheless, some clarifications and further justifications are required to further advance the quality of this work, which you can find in detail, along with some editorial requests, at the end of this letter.

After discussing the paper with the editorial team and an academic editor with relevant expertise, I’m pleased to invite you to revise the paper in response to the reviewers’ comments. We plan to send the revised paper to some or all of the original reviewers, and we cannot provide any guarantees at this stage regarding publication. Please note that in the next round of revision we might need to recruit additional reviewers.

We ask that you submit your revision by Feb 23 2026 11:59PM. However, if this deadline is not feasible, please contact me by email, and we can discuss a suitable alternative.

Don’t hesitate to contact me directly with any questions (efourli@plos.org).

Best regards,

Evangelia

Evangelia Fourli, Ph.D.

Associate Editor

PLOS Medicine

efourli@plos.org

Comments from the reviewers:

Reviewer #1: This sentence in the "Results" section is confusing and unclear: First contact and early sorting of children into these pathways frequently involved guards, lay bystanders, students, and caregivers, with duplicated assessments when assessment and treatment occurred in different locations and misrouting of patients away from intended pathways. Consider making it two sentences.

The acronym "FGD" in Design/Methods is not previously defined.

The references are included twice - pages 22-25 and again on pages 67-69

Reviewer #2: This is a well-conducted process mapping study on the pathways of emergency care for severely ill children in Nigerian and Ugandan hospitals. This is basically a qualitative, multi-method study based on FGDs, therefore not much of statistical aspects that I can comment on as a statistical reviewer. However, there are still a few issues needing attention.

1. It says in the abstract 'We conducted a qualitative, multi-method study, across 26 purposively selected secondary and tertiary facilities in Uganda and Nigeria between October 2023 to December 2024'. However, there is no explicit justification on the selection of these 26 facilities for analyses. How representative are these facilities for Nigerian and Ugandan hospitals as a whole?

2. There are no explicit annotations for abbreviations such as FGDs and ICA. It is difficult to follow for readers who are not in the field.

3. An expert with knowledge and experience of the process mapping tool is needed to thoroughly review the analyses results.

Reviewer #3: Thank you for the privilege of reviewing this manuscript. I would like to commend the authors for this work. Overall, this is a strong, thoughtful, and policy-relevant qualitative study that addresses a critical but under-described gap in paediatric emergency care in low-resource settings, that is, how care is actually organised and delivered in real-world facilities. The use of prolonged facility embedding, process mapping, and cross-country comparison is a major strength, and the proposed organizational "archetypes" provide a useful and transferable conceptual framework. The manuscript is well written and methodologically rigorous. I believe with some adjustments/revisions we could have a much improved paper.

1. The manuscript clearly positions itself alongside ETAT and other standardized/guideline-based emergency care approaches, but the added conceptual value of the archetype framework could be articulated more explicitly and earlier. While this is addressed well in the discussion, the introduction could more clearly state why existing frameworks are insufficient for understanding real-world organizational care and how archetypes complement, rather than compete with, guideline-based approaches.

A short paragraph explicitly stating what this study enables, whereas previous facility readiness assessments or intervention studies do not, would strengthen the framing.

2. The four archetypes look intuitively appealing and seem well supported by data, but how stable are they over time, with evolving HR and logistical ecosystems in SSA? And are facilities expected to fit one archetype neatly, or are hybrids anticipated?

I suggest adding a brief clarification (perhaps in the discussion) that archetypes are heuristic tools rather than rigid categories and that facilities may shift between them depending on staffing, time of day, or resource availability. This would pre-empt over-interpretation and enhance transferability.

3. One of the most compelling findings is the central role played by guards, caregivers, students, and bystanders in triage and navigation (this is also my lived reality). Now, while this is well described, the implications are not fully unpacked. Questions that quickly come to mind are: To what extent should health systems formalise these roles? And where do the authors see the balance between legitimising informal adaptations and safeguarding quality and accountability?

Even a short reflective paragraph acknowledging these tensions would strengthen the manuscript and avoid the perception that the paper implicitly endorses all informal practices. (To avoid the pitfalls of task shifting as a policy)

4. While neonatal care pathways are discussed separately and somewhat briefly, are they intended as a secondary analytical strand or illustrative extensions of the archetype framework?

The authors can attempt to explicitly acknowledge the limits of neonatal data earlier in the Results section, rather than mainly in Methods/Limitations. If not possible, then a little more can be said in the discussion (limitations)

5. The description of inductive content analysis is clear, but the transition from codes to archetypes could benefit from a short, worked example. Consider a simple schematic or table in the main paper (not appendix) showing how observations led to archetype construction.

6. While terms such as "OPD," "IPD," and "ED" are well defined (and maybe well understood by SSA readers), for a global readership, occasional reminders (especially early in the Results) would help. Furthermore, please consider briefly defining "critical realist ontology" in plain language for non-qualitative specialists.

7. While differences between countries are evident, the paper appropriately avoids over-comparison. However, one explicit statement clarifying that the archetypes cut across national boundaries (rather than being country-specific models) would be helpful.

8. The swimlane diagrams are excellent, but to me, they seem information-dense. Is it possible to have one simplified archetype diagram for the main text, with the others remaining in the appendix, to improve readability? For many, such an arrangement may be a relatively new graphical concept.

9. The conclusions are appropriate and measured. I believe one final sentence explicitly linking findings to national emergency care planning or hospital accreditation processes could enhance policy relevance.

Dr Arthur Kwizera

Reviewer #4: Comments on the manuscript "Pathways of emergency care for severely ill children in Nigerian and Ugandan hospitals: a process mapping study"

The multi-site study described processes in the delivery of emergency care in Nigeria and Uganda, two resource-limited countries with high child and neonatal mortalities. The study purposely selected smaller and more remote secondary and tertiary hospitals as these were underrepresented in literature. Using a mapping process and FGDs with researchers who were embedded in these facilities, the study revealed a variety of gaps - both inherent in the service delivery process and in the administrative aspect - that undermined the timely and efficient provision of emergency care. The findings highlight actionable areas that can help improve systems and eventually mitigate child deaths.

MY COMMENTS

The paper is well written and the main points are clearly articulated.

The methodology is rigorous and detailed in description.

The findings are robust and well documented.

The discussion is sufficient for the aim of the paper.

Nonetheless, there are areas that can still be improved.

The Abstract and the manuscript itself is quite lengthy. The language is not verbose but there are descriptions in the Methodology that can be edited to shorten the paper.

The study facilities can do with more description aside from where they are and if they are secondary or tertiary hospitals. For instance, are these government or privately-run facilities? Are they under local government or national government supervision? Some of these data are in Table 1 but a line or two in this regard will help. Such information can help clarify the funding, staffing, and governance aspects later in the Discussion.

Similarly, it may not be enough to state that the researchers were "embedded" in the facilities. In studies with FGDs, a description of the study participants, even briefly (or even in the form of a table), is provided. The "Researcher Positionality" in the appendix does not suffice. For instance, how long have these researchers been working with their respective facility? This information will strengthen the reliability of their comments, especially if most of them have been with their facility for a long time.

The term "cost-cutting opportunities" in the Conclusion section should be reconsidered as it may be considered a misnomer. For one, cost-cutting measures are usually not regarded as opportunities. More importantly, in resource-poor or resource-limited settings, cost-cutting will only erode whatever is left that people are already hard-pressed to work with. Perhaps the term "cost-efficient measures" may be an alternative.

Corollary to the aforementioned point, the proposed "cost-cutting opportunities" in the Conclusion section should be fleshed out in a separate paragraph in the Discussion.

Overall, this paper can be accepted for publication, but can still be improved with some editing and very minor revisions.

---

* Please upload any figures associated with your paper as individual TIF or EPS files with 300dpi resolution at resubmission; please read our figure guidelines for more information on our requirements: http://journals.plos.org/plosmedicine/s/figures. While revising your submission, we strongly recommend that you use PLOS's NAAS tool (https://ngplosjournals.pagemajik.ai/artanalysis) to test your figure files. NAAS can convert your figure files to the TIFF file type and meet basic requirements (such as print size, resolution), or provide you with a report on issues that do not meet our requirements and that NAAS cannot fix.

After uploading your figures to PLOS's NAAS tool - https://ngplosjournals.pagemajik.ai/artanalysis, NAAS will process the files provided and display the results in the "Uploaded Files" section of the page as the processing is complete.

If the uploaded figures meet our requirements (or NAAS is able to fix the files to meet our requirements), the figure will be marked as "fixed" above. If NAAS is unable to fix the files, a red "failed" label will appear above.

When NAAS has confirmed that the figure files meet our requirements, please download the file via the download option, and include these NAAS processed figure files when submitting your revised manuscript.

FIGURES AND TABLES

SUPPLEMENTARY MATERIAL

REFERENCES

QUALITATIVE STUDIES

* Please report your qualitative study according to the appropriate study design provided at (http://www.equator-network.org/?post_type=eq_guidelines&eq_guidelines_study_design=qualitative-research&eq_guidelines_clinical_specialty=0&eq_guidelines_report_section=0&s=) and provide the relevant completed checklist as a supplemental file. In the checklist, please include sufficient text excerpted from the manuscript to explain how you accomplished all applicable items. When completing checklists, please use section and paragraph numbers, rather than page numbers.

* We recommend that authors use the COREQ checklist, or other relevant checklists listed by the Equator Network, such as the SRQR, to ensure complete reporting (see: http://www.equator-network.org/?post_type=eq_guidelines&eq_guidelines_study_design=qualitative-research&eq_guidelines_clinical_specialty=0&eq_guidelines_report_section=0&s=). Please add the following statement, or similar, to the Methods: "This study is reported as per the Consolidated criteria for reporting qualitative research (COREQ): a 32-item checklist for interviews and focus groups (S1 Checklist)."

* In general, we expect qualitative studies to include the following: 1) defined objectives or research questions; 2) description of the sampling strategy, including rationale for the recruitment method, participant inclusion/exclusion criteria and the number of participants recruited; 3) detailed reporting of the data collection procedures; 4) data analysis procedures described in sufficient detail to enable replication; 5) a discussion of potential sources of bias; and 6) a discussion of limitations.

---

## [Decision Letter · Decision Letter 2]

14 Apr 2026

Dear Dr. Subhi,

Thank you very much for re-submitting your manuscript "Pathways of emergency care for severely ill children in Nigerian and Ugandan hospitals: a process mapping study." (PMEDICINE-D-25-03978R2) for review by PLOS Medicine.

I have discussed the paper with my colleagues and the academic editor and it was also seen again by 2 reviewers. I am pleased to say that provided the remaining editorial and production issues are dealt with we are planning to accept the paper for publication in the journal.

[LINK]

We look forward to receiving the revised manuscript by Apr 20 2026 11:59PM.

Sincerely,

Evangelia Fourli, Ph.D.

Senior Editor

PLOS Medicine

plosmedicine.org

Requests from Editors:

Please note that not all of the following comments apply to your manuscript.

GENERAL EDITORIAL REQUESTS

"* At this stage, we ask that you include a short, non-technical Author Summary of your research to make findings accessible to a wide audience that includes both scientists and non-scientists. The Author Summary should immediately follow the Abstract in your revised manuscript. This text is subject to editorial change and should be distinct from the scientific abstract. Ideally each sub-heading should contain 2-3 single sentence, concise bullet points containing the most salient points from your study. In the final bullet point of ‘What Do These Findings Mean?’ Please include the main limitations of the study in non-technical language.

Please see our author guidelines for more information: https://journals.plos.org/plosmedicine/s/revising-your-manuscript#loc-author-summary."

* Please confirm that your title complies with PLOS Medicine's style. Your title must be nondeclarative and not a question. It should begin with main concept if possible. "Effect of" should be used only if causality can be inferred, i.e., for an RCT. Please place the study design ("A randomized controlled trial," "A retrospective study," "A modelling study," etc.) in the subtitle (ie, after a colon).

* Please confirm that your abstract complies with our requirements, including format (three sections: Background, Methods and Findings, and Conclusions) and providing all the information relevant to this study type https://journals.plos.org/plosmedicine/s/submission-guidelines#loc-abstract

* Please ensure that the Introduction ends with a clear description of the study question or hypothesis.

* Please ensure that all abbreviations are defined at first use throughout the text.

* Please confirm that all numbers presented in the abstract are present and identical to numbers presented in the main manuscript text.

GENERAL

* Please review your text for claims of novelty or primacy (e.g. 'for the first time') and remove this language. In addition, please check that any use of statistical terms (such as trend or significant) are supported by the data, and if not please remove them.

* Please remove the 'conclusions' subheading from the discussion. Please also remove any other subheadings from the discussion. There should be no conclusions section. Discussion should only be referred to as such and no subheadings should be present.

* In the author summary, in the final bullet point of 'What Do These Findings Mean?', please include the main limitations of the study in non-technical language. Also, please adhere to the the three main sections:

"-Why Was This Study Done?

-What Did the Researchers Do and Find?

-What Do These Findings Mean?"

* Please include an Acknowledgments section in your manuscript.

* Please report your qualitative study according to the appropriate study design provided at http://www.equator-network.org/?post_type=eq_guidelines&eq_guidelines_study_design=qualitative-research&eq_guidelines_clinical_specialty=0&eq_guidelines_report_section=0&s= and provide the relevant completed checklist. In the checklist please include sufficient text excerpted from the manuscript to explain how you accomplished all applicable items.

* Please describe the conceptual framework underlying your qualitative analysis.

* Please revise the text, as sentences are sometimes too dense or unclear (example paragraph 1 of the introduction, line 499, etc).

* Please remove all highlighted text.

FUNDING STATEMENT

* The funding statement should include: specific grant numbers, initials of authors who received each award, URLs to sponsors’ websites. Also, please state whether any sponsors or funders (other than the named authors) played any role in study design, data collection and analysis, the decision to publish, or preparation of the manuscript. If they had no role in the research, include this sentence: “The funders had no role in study design, data collection and analysis, decision to publish, or preparation of the manuscript.”

COMPETING INTERESTS STATEMENT

* All authors must declare their relevant competing interests per the PLOS policy, which can be seen here: https://journals.plos.org/plosmedicine/s/competing-interests For authors with ties to industry, please indicate whether any of the interests has a financial stake in the results of the current study.

FIGURES and TABLES

*For figure S1, can you please substitute the handwritten maps with computer-made ones? It may be hard for the reader to comprehend the images. Please also note that there is a misalignment between figure letters (A,B,C) and the images.

*In supplementary material please update the contents to reflect exactly what is shown, ie "Table S1. Deductive codes, following steps in the patient journey through a health facility, used to organise the data." Also, please remove table and figure titles and keep only the legends, which already include the title.

*For tables please place legends above the table, while for figures below the relevant figure.

*Please, also update the order of the contents in supplementary material (and the corresponding references in the main text) as stated in the manuscript:

"Supporting information

Appendix S1

Table S1 – Deductive codes, following steps in the patient journey through a health facility, used to organise the data.

Table S2. Description of facilities that align with Archetype A

Table S3. Description of facilities that align with Archetype B

Table S4. Description of facilities that align with Archetype C

Table S5. Description of facilities that align with Archetype D

Table S6. Initial assessment, treatment and admission locations for neonates (<29 days)

Table S7. Management of ‘special’ populations: older children, trauma and surgical

conditions

Table S8. Quotes from focus group discussions with researchers, organised under categories that explain steps in the patient journey through a facility

Table S9. Characteristics of researchers participating in focus group discussions in Uganda and Nigeria

Figure S1. Example of the evolution of process maps over time

Text. Researcher positionality

Text. Process mapping data collection tool

Text. Focus group discussions"

Comments from Reviewers:

Reviewer #2: Thanks authors for their effort to improve the manuscript. I am satisfied with the response and revision. No further issues needing attention.

Reviewer #4: I am satisfied with the manner by which the authors have addressed my concerns. The revised manuscript is quite satisfactory.

[LINK]

---

## [Editor Report · Decision Letter 3]

27 Apr 2026

Dear Dr Subhi,

On behalf of my colleagues and the Academic Editor, Dr Margaret E Kruk, I am pleased to inform you that we have agreed to publish your manuscript "Pathways of emergency care for severely ill children in Nigerian and Ugandan hospitals: a process mapping study." (PMEDICINE-D-25-03978R3) in PLOS Medicine.

PRESS

Sincerely,

Evangelia Fourli, Ph.D.

Associate Editor

PLOS Medicine